# Approximate Inference Suffices for Statistical Distance Estimation

## Abstract

Statistical distance (also known as total variation distance) and probabilistic inference are fundamental notions, widely used in machine learning, information theory, and high-dimensional statistics. While there are efficient algorithms that can estimate statistical distance or probabilistic inference in some specific settings, it has remained an open problem to see whether these two notions can be approximately reduced to each other. In this work, we take the first step in addressing this problem, and show that estimating statistical distance can be reduced to estimating probabilistic inference, via an efficient *structure preserving* randomized reduction. This allows us to use approximate inference algorithms to multiplicatively estimate statistical distance in directed graphical models.

## 1 Introduction

Probabilistic inference and statistical distance estimation are two fundamental computational problems that lie at the heart of machine learning, statistics, and artificial intelligence. While both problems are known to be #P-complete in their exact forms, they have been extensively studied from different algorithmic perspectives, leading to rich bodies of work on approximation algorithms for each problem independently. A natural and important question is whether these two central problems can be computationally reduced to each other, potentially allowing algorithmic advances in one area to benefit the other.

**Statistical Distance.** The statistical distance (also known as total variation distance or TV distance) between distributions $P$ and $Q$ over a common sample space $D$, denoted by $d_{\text{TV}}(P, Q)$, is defined as

$$d_{\text{TV}}(P, Q) := \frac{1}{2} \sum_{x \in D} |P(x) - Q(x)|.$$

Statistical distance is a fundamental measure in probability theory and statistics, possessing many desirable mathematical properties: it is bounded in $[0, 1]$, forms a metric, is invariant under bijections, and has a natural probabilistic interpretation as the maximum gap between probabilities assigned to any event by the two distributions. These properties make it a versatile and widely-used distance measure across diverse areas including machine learning (Shalev-Shwartz & Ben-David, 2014), information theory (Cover & Thomas, 2006), cryptography (Stinson, 1995), differential privacy (Dwork, 2006), and pseudorandomness (Vadhan, 2012).

The computational complexity of statistical distance has attracted significant attention. Sahai and Vadhan (Sahai & Vadhan, 2003) established that additively approximating statistical distance between circuit-samplable distributions is complete for Statistical Zero Knowledge (SZK). Subsequent work has explored variations of this theme (Goldreich et al., 1999; Malka, 2015; Dixon et al., 2020), showing, for example, that deciding closeness to uniform is complete for Non-Interactive Statistical Zero Knowledge (NISZK). Additionally, several works have demonstrated that statistical distance computation for hidden Markov models is undecidable and #P-hard to approximate (Cortes et al., 2007; Lyngsø & Pedersen, 2002; Kiefer, 2018). Most relevant to our work, Bhattacharyya et al. (2023) proved that exactly computing statistical distance between product distributions is #P-complete, and multiplicatively approximating it for Bayes nets is NP-hard.

On the algorithmic side, there has been substantial progress on efficient approximation. Bhattacharyya et al. (2020) designed algorithms for additively approximating statistical distance for structured

distributions including Bayes nets, Ising models, and multivariate Gaussians. For multiplicative approximation, recent breakthroughs include FPTASes for product distributions (Bhattacharyya et al., 2023; Feng et al., 2023; 2024), an FPRAS for Bayes nets of small treewidth (Bhattacharyya et al., 2024), algorithms for Gaussian distributions (Bhattacharyya et al., 2025), and algorithms for spin systems (Feng et al., 2025). Interestingly, Feng et al. (2025) show a reduction from approximating statistical distance to approximate counting, albeit for a *restricted* class of spin systems. In our work, we give a broader reduction (Theorem 1) from approximating statistical distance to approximate counting, in the context of Bayes nets.

**Probabilistic Inference.** Probabilistic inference in graphical models is equally fundamental, with applications spanning statistics, machine learning, and artificial intelligence (e.g., see (Wainwright et al., 2008)). The problem involves computing marginal probabilities of the form $\mathbf{Pr}[X_1 \in S_1, \ldots, X_n \in S_n]$ for random variables $X_1, \ldots, X_n$ in a graphical model and sets $S_1, \ldots, S_n$.

Exact algorithms for probabilistic inference include message passing (Pearl, 1988), variable elimination (Dechter, 1999), and junction-tree propagation (Lauritzen & Spiegelhalter, 1988). However, the problem is #P-complete in general (Cooper, 1990; Littman et al., 2001; Roth, 1996), motivating extensive research on approximate methods. These include loopy belief propagation, variational inference (Wainwright et al., 2008), and particle-based algorithms (see Chapter 13 of (Koller & Friedman, 2009)). The rich landscape of approximate inference techniques represents decades of algorithmic development and is a cornerstone of modern probabilistic machine learning.

**The Fundamental Gap and Our Contribution.** Despite being central problems in computational probability, the relationship between statistical distance estimation and probabilistic inference has remained largely unexplored. This represents a significant gap in our understanding, particularly given that both problems are #P-complete and both admit sophisticated approximation algorithms.

Recently, Bhattacharyya et al. (2024) took the first step toward bridging this gap by showing that statistical distance between Bayes nets can be estimated using *exact* probabilistic inference queries. While this was an important theoretical advance, it is practically limiting: Exact probabilistic inference is #P-complete and thus computationally intractable for most realistic instances. This limitation severely restricts the applicability of their approach, since practitioners must rely on the limited cases where exact inference is feasible, forgoing the rich ecosystem of approximate inference algorithms that make probabilistic modeling tractable in practice.

In this work, we resolve this fundamental limitation by establishing the first reduction from statistical distance estimation to *approximate* probabilistic inference, formally defined as follows. Given a Bayes net over random variables $X_1, \ldots, X_n$, alphabet $\Sigma$, and sets $S_1, \ldots, S_n \subseteq \Sigma$, and a parameter $\varepsilon$, compute a number $p$ such that

$$\mathbf{Pr}[X_1 \in S_1, \ldots, X_n \in S_n] / (1 + \varepsilon) \le p \le \mathbf{Pr}[X_1 \in S_1, \ldots, X_n \in S_n] (1 + \varepsilon).$$

**Theorem 1.** *There is an FPRAS for estimating statistical distance between Bayes nets of alphabet $[\ell]$ over the same underlying directed acyclic graph on $n$ nodes, that uses* approximate *probabilistic inference queries. The running time of this FPRAS is $O(n^3 \ell \varepsilon^{-2} \log \delta^{-1})$, where $\varepsilon, \delta$ are the accuracy and confidence error parameters, respectively.*

Our reduction immediately makes the extensive toolkit of approximate inference algorithms—including variational methods, sampling techniques, and message passing algorithms—applicable to statistical distance problems. This dramatically expands the practical reach of statistical distance estimation for complex, high-dimensional distributions.

**Settings Where Approximate Inference Is Easy but Exact Inference Is Hard.** Approximate inference is a well-studied topic in Bayesian networks with a long series of results from theory as well as practice. Practical methods such as variational inference, belief propagation have been shown to converge even when the exact inference is intractable in practice. Theoretical results have often shown tractability under conditions on the CPT parameters Ng & Jordan (1999); Saul & Jordan (2013); Kearns & Saul (2011); Jordan et al. (1999) — these methods have also shown to converge in practice.

The practical implication of our result is that for the classes of networks for which approximation techniques such as variational inference and belief propagation converge, it is also possible to approximate the total variation distance.

## 1.1 PAPER ORGANIZATION

We present some background material in Section 2. Theorem 1 is proved in Section 3.

## 2 PRELIMINARIES

We require the following definition of approximation algorithms.

**Definition 2** (FPRAS). A function $f : \{0,1\}^* \to \mathbb{R}$ admits a *fully polynomial-time randomized approximation scheme (FPRAS)* if there is a *randomized* algorithm $\mathcal{A}$ such that for all $n$ and all inputs $x \in \{0,1\}^*$, $\varepsilon > 0$, and $\delta > 0$, the algorithm $\mathcal{A}$ outputs a $(1+\varepsilon)$-multiplicative approximation of $f(x)$, i.e., a value that lies in the interval $[f(x)/(1+\varepsilon), (1+\varepsilon)f(x)]$, with probability $1 - \delta$. The running time of $\mathcal{A}$ is polynomial in $|x|, 1/\varepsilon, \log(1/\delta)$.

We shall also use the following in the proof of Theorem 1.

**Lemma 3** (Hoeffding's inequality). *Let $X_1, \ldots, X_n$ be independent random variables such that $a_i \leq X_i \leq b_i$ for all $1 \leq i \leq n$. Then*

$$\mathbf{Pr}\left[\left|\sum_{i=1}^n X_i - \mathbf{E}\left[\sum_{i=1}^n X_i\right]\right| \geq t\right] \leq 2 \exp\left(-2t^2 / \sum_{i=1}^n (b_i - a_i)^2\right).$$

## 2.1 BAYES NETS

We formally define Bayes nets. For a directed acyclic graph (DAG) $G$ and a node $v$ in $G$, let $\Pi(v)$ denote the set of parents of $v$.

**Definition 4** (Bayes Nets). A *Bayes net* is specified by a DAG over a node set $[n]$ and a collection of probability distributions, as follows. Each node $i$ is associated with a random variable $X_i$ whose range is an alphabet $[\ell]$ (for some $\ell > 0$) and has a Conditional Probability Table (CPT) that describes the following: For every $x \in [\ell]$ and every $y \in [\ell]^k$, whereby $k = |\Pi(i)|$, the CPT has the value of $\mathbf{Pr}[X_i = x | X_{\Pi(i)} = y]$ stored (which is a finite, rational number). Given such a Bayes net, its associated probability distribution $P$ is given as follows. Let us denote by $X$ the joint distribution $(X_1, \ldots, X_n)$ and $x_{\Pi(i)}$ the projection of $x$ to $\Pi(i)$. Then for all $x \in [\ell]^n$, the probability $P(x)$ is equal to $\mathbf{Pr}_P[X = x] = \prod_{i=1}^n \mathbf{Pr}_P[X_i = x_i | X_{\Pi(i)} = x_{\Pi(i)}]$.

## 2.2 PRIOR WORK ON STATISTICAL DISTANCE ESTIMATION

The recent work of Bhattacharyya et al. (2024) provides an FPRAS for statistical distance estimation using exact probabilistic inference oracles. Their approach employs the classical importance sampling technique. Informally, in the simplest case, importance sampling works as follows. Suppose the goal is to estimate the probability of an event $A$ over a sample space $\Omega$. If this probability is very small, then the standard Monte Carlo approach is unlikely to yield a good multiplicative approximation, since most samples will not fall in $A$. The importance sampling technique defines an alternative, efficiently samplable distribution $\pi$, under which the probability of $A$ is significantly larger. We then estimate the probability of $A$ by sampling from $\pi$, and reweighting the probabilities appropriately, typically by multiplying by the ratio between the original distribution and $\pi$.

The key insight of Bhattacharyya et al. (2024) is that they define an estimator function $f$ and a distribution $\pi$ such that $\mathbf{E}_\pi[f] = d_{\mathrm{TV}}(P, Q)/Z$ where $Z$ is a normalization constant (as described below) that can be computed efficiently.

Since $d_{\mathrm{TV}}(P, Q) = \sum_w g^*(w)$ where $g^*(w) := P(w) - \min(P(w), Q(w))$, the challenge is to construct a tractable surrogate function $g$ for $g^*(w)$ that can be used for importance sampling. The key technical contribution is the introduction of a *partial coupling* approach. Rather than working directly with $\min(P(w), Q(w))$, which is the minimum of products, the prior work defines

$$h(w, i) := \min\left(P_{i|\Pi(i)}\big(w_i | w_{\Pi(i)}\big), Q_{i|\Pi(i)}\big(w_i | w_{\Pi(i)}\big)\right),$$

and sets $h(w) := \prod_{i=1}^{n} h(w,i)$ and $g(w) := P(w) - h(w)$, which provides a tractable upper bound on $g^*(w)$ since $h(w) \leq \min(P(w), Q(w))$. The estimator function is then defined as $f(w) = g^*(w)/g(w)$ (note that $0 \leq f(w) \leq 1$), and the importance sampling distribution is $\pi(w) = g(w)/Z$ where $Z = \sum_w g(w)$ is a normalization constant.

To enable computation of $Z$ and sampling from $\pi$, the prior work constructs an auxiliary Bayes net distribution over alphabet $[\ell]^2$, representing a joint distribution $(X, Y)$ where $X$ and $Y$ take values over $[\ell]^n$. The CPT for $(X, Y)$ is carefully designed so that for nodes where both $X_i$ and $Y_i$ take the same value $b$:

$$\mathbf{Pr}\big[(X_i, Y_i) = (b, b) \,|\, \big(X_{\Pi(i)}, Y_{\Pi(i)}\big) = (c_1, c_2)\big] = \min\big(P_{i|\Pi(i)}(b|c_1), Q_{i|\Pi(i)}(b|c_2)\big),$$

and the remaining probabilities are set to ensure that the marginal $X$ follows distribution $P$. This can be done as follows. For every $z \neq b$ set

$$\mathbf{Pr}\big[(X_i, Y_i) = (b, z) \,|\, \big(X_{\Pi(i)}, Y_{\Pi(i)}\big) = (c_1, c_2)\big]$$
$$:= \frac{P_{i|\Pi(i)}(b|c_1) - \min\big(P_{i|\Pi(i)}(b|c_1), Q_{i|\Pi(i)}(b|c_2)\big)}{\ell - 1}.$$

The prior algorithm computes $Z$ exactly, and samples from $\pi$ by iteratively sampling each coordinate using exact conditional probabilities. Specifically, after sampling $(w_1, \ldots, w_{k-1})$, the next coordinate $w_k$ is sampled with probability proportional to $Z_{b_1, \ldots, b_{k-1}, b}$ for each $b \in [\ell]$, where $Z_{b_1, \ldots, b_k}$ represents the normalization constant restricted to assignments starting with $(b_1, \ldots, b_k)$.

**Why Exact Probabilistic Inference Was Necessary.** The crucial observation is that $Z = \mathbf{Pr}[X \neq Y]$ under this joint distribution.

**Claim 5.** $Z = \mathbf{Pr}[X \neq Y]$

To see this, note that since $X \sim P$ and for that matter $P(w) = \mathbf{Pr}[X = w]$, we have

$$g(w) = P(w) - \prod_{i=1}^{n} \min\big(P_{i|\Pi(i)}\big(w_i|w_{\Pi(i)}\big), Q_{i|\Pi(i)}\big(w_i|w_{\Pi(i)}\big)\big)$$
$$= P(w) - \mathbf{Pr}[X = Y = w]$$
$$= \mathbf{Pr}[X = w] - \mathbf{Pr}[X = Y = w]$$
$$= \mathbf{Pr}[X = w] - \mathbf{Pr}[Y = w|X = w] \cdot \mathbf{Pr}[X = w]$$
$$= \mathbf{Pr}[X = w]\,(1 - \mathbf{Pr}[Y = w|X = w])$$
$$= \mathbf{Pr}[X = w]\,\mathbf{Pr}[Y \neq w|X = w] = \mathbf{Pr}[X = w, Y \neq w].$$

Therefore, $Z = \sum_w g(w) = \mathbf{Pr}[X \neq Y]$. The above can be extended to show that $Z_{b_1 \cdots b_k} = \sum_{w:(w_1, \ldots, w_k)=(b_1, \ldots, b_k)} g(w)$. Now, while it might seem natural to approximate $Z = 1 - \mathbf{Pr}[X = Y]$ using an approximate inference oracle to compute $\mathbf{Pr}[X = Y]$, this approach fails because if $\widetilde{\mathbf{Pr}[X = Y]}$ is an $(1 + \varepsilon)$-approximation of $\mathbf{Pr}[X = Y]$, then $1 - \widetilde{\mathbf{Pr}[X = Y]}$ can have arbitrarily large relative error when $\mathbf{Pr}[X = Y]$ is close to 1.

This fundamental limitation necessitated the use of exact probabilistic inference oracles in the original work. The correctness of the prior approach relies critically on the exact computation of $Z$ and the conditional probabilities $Z_{b_1, \ldots, b_k}$, which is the main limitation we address in this work.

## 3 FROM TOTAL VARIATION DISTANCE TO PROBABILISTIC INFERENCE

Our main contribution is an algorithm that estimates total variation distance using only *approximate* probabilistic inference queries. We build on the importance sampling framework where:

- An auxiliary Bayes net $\mathcal{L}$ is constructed over $G$ with alphabet $[\ell]^2$ representing a joint distribution $(X, Y)$ where $X \sim P$;
- An estimator function $f$ and distribution $\pi$ are defined such that $d_{\text{TV}}(P, Q) = Z \cdot \mathbb{E}_{w \sim \pi}[f(w)]$ for a normalization constant $Z$.

The algorithm has three primary components:

1. **Normalization estimation**: Estimate the normalization constant $Z$ and related quantities using approximate inference queries.

2. **Approximate sampling**: Sample from an approximate version $\widetilde{\pi}$ of the target distribution $\pi$, using approximate probabilistic inference queries.

3. **Monte Carlo estimation**: Combine samples to estimate $d_{\mathrm{TV}}(P, Q)$.

We present our main algorithm for estimating total variation distance using approximate probabilistic inference. See Algorithm 1. In this algorithm the procedure ESTIMATENORMALIZATION returns an approximate value of $Z$. The procedure APPROXIMATESAMPLE returns an approximate sample from the distribution $\pi$. The pseudocode for these procedures are described in Algorithm 2 and Algorithm 3.

---

**Algorithm 1** FPRAS for $d_{\mathrm{TV}}$ estimation via approximate probabilistic inference.

---

**Require:** Bayes nets $P, Q$ over DAG $G$ with $n$ nodes, parameters $\varepsilon, \delta$
**Ensure:** $(1 + \varepsilon)$-approximation of $d_{\mathrm{TV}}(P, Q)$ with probability $1 - \delta$
1: Construct the auxiliary Bayes net $\mathcal{L}$ over $G$
2: $\widetilde{Z} \leftarrow$ ESTIMATENORMALIZATION$(P, Q, (), \varepsilon/3, \delta/3)$          {*Empty prefix.*}
3: **if** $\widetilde{Z} = 0$ **then**
4:     **return** $0$
5: $m \leftarrow Cn^2\varepsilon^{-2}\log\delta^{-1}$ for a sufficiently large constant $C$
6: $F \leftarrow 0$
7: **for** $i \leftarrow 1, \ldots, m$ **do**
8:     $w^i \leftarrow$ APPROXIMATESAMPLE$(P, Q, \varepsilon/3, \delta/(3m))$
9:     $F \leftarrow F + f(w^i)$          {*Compute estimator function.*}
10: $\widetilde{d_{\mathrm{TV}}} \leftarrow \widetilde{Z} \cdot F/m$
11: **return** $\widetilde{d_{\mathrm{TV}}}$

---

### 3.1 ESTIMATING NORMALIZATION CONSTANTS

Though it suffices to estimate $Z$ in Algorithm 1, we need to estimate a more generalized term . Given $b_1 \cdots b_k \in [\ell]^k$, let $Z_{b_1,\ldots,b_k} = \mathbf{Pr}[X \neq Y, X_1 = b_1, \ldots, X_k = b_k]$. Note that the unconditioned case $Z = \mathbf{Pr}[X \neq Y]$ corresponds to $k = 0$. The Algorithm 2 estimates $Z_{b_1,\ldots,b_k}$. First observe that we can express $Z_{b_1,\ldots,b_k} = \mathbf{Pr}[\bigcup_{i=1}^n (X_i \neq Y_i, X_1 = b_1, \ldots, X_k = b_k)]$. Let us denote $E_i$ as the event $(X_i \neq Y_i, X_1 = b_1, \ldots, X_k = b_k)$ and therefore, we are interested in estimating $\mathbf{Pr}[\cup_i E_i]$. The algorithm works in two steps:

**Step 1: Estimate probabilities of individual events:** For event $E_i$ note that

$$\mathbf{Pr}[E_i] = \mathbf{Pr}[X_i \neq Y_i, X_1 = b_1, \ldots, X_k = b_k] = \mathbf{Pr}[(X_1, Y_1) \in S_1, \ldots, (X_n, Y_n) \in S_n].$$

where for $j \leq k$ such that $j \neq i$, we have $S_j = \{b_j\} \times [\ell]$, and $S_i = \{b_i\} \times ([\ell] \setminus \{b_i\})$. Moreover, in this case, $S_{k+1} = \cdots = S_n = [\ell]^2$. Thus, one call to the approximate probability inference oracle gives an estimate of $\mathbf{Pr}[E_i]$.

**Step 2: Monte Carlo estimation:** The goal of this step is to estimate the $\mathbf{Pr}[\cup_i E_i]$. For this, we first pick an event $E_i$ with its proportional (estimated) probability (Line 21). Then we sample a string $z$ from the event $E_i$ (Lines 22–27). Then check if the sample $z$ does not appear in earlier events $E_1, \ldots E_{i-1}$, and if so, increment a counter $c$ (Lines 28–32). This avoids oversampling. We now explain how to sample $z$ from $E_i$. To sample from event $E_i$, we use conditional sampling (in Lines 25–27) to sequentially build an $z$ that is in event $E_i$. Suppose we have built a prefix of $z$, the next step is to append $z$. For every possible extension $(s_1, s_2) \in [\ell]^2$ of $z$, we estimate the probability a sample from $E_i$ starts with of $(z(s_1, s_2))$. Note that this can be done by making a call to an approximate inference oracle. If $\widetilde{v_{s^{(1)}, s^{(2)}}}$ is the estimate probability, then we extend $z$ with $(s_1, s_2)$ with probability $\widetilde{v_{s_1, s_2}} / \sum_{s^{(1)}, s^{(2)} \in [\ell]} \widetilde{v_{s^{(1)}, s^{(2)}}}$.

---

**Algorithm 2** ESTIMATENORMALIZATION: An FPRAS for estimating $Z_{b_1,\ldots,b_k}$ via approximate probabilistic inference.

---

**Require:** Bayes nets $P, Q$ over DAG with $n$ nodes, alphabet $[\ell]$, prefix constraints $(b_1, \ldots, b_k)$, parameters $\varepsilon, \delta$.
**Ensure:** $(1 + \varepsilon)$-approximation of $Z_{b_1,\ldots,b_k}$ with probability $1 - \delta$.
1: {**Step 1: Estimate weights of individual events**}
2: **for** $i \leftarrow 1, \ldots, n$ **do**
3:   **if** $i \leq k$ **then**
4:     Query for $\mathbf{Pr}[(X_1, Y_1) \in S_1, \ldots, (X_n, Y_n) \in S_n]$ where:
5:       $S_i = \{b_i\} \times ([\ell] \setminus \{b_i\})$                         $\{X_i = b_i \neq Y_i\}$
6:       $S_j = \{b_j\} \times [\ell]$ for $j < i, j \leq k$              $\{X_j = b_j\}$
7:       $S_j = [\ell]^2$ for $j > k$
8:     $\widetilde{w_i} \leftarrow$ query result
9:   **else**
10:     $\widetilde{w_i} \leftarrow 0$
11:     **for** $b \leftarrow 1, \ldots, \ell$ **do**
12:       Query for $\mathbf{Pr}[(X_1, Y_1) \in S_1, \ldots, (X_n, Y_n) \in S_n]$ where:
13:         $S_j = \{b_j\} \times [\ell]$ for $j \leq k$                     $\{X_j = b_j\}$
14:         $S_i = \{b\} \times ([\ell] \setminus \{b\})$                   $\{X_i = b \neq Y_i\}$
15:         $S_j = [\ell]^2$ for $j \neq i, j > k$
16:       $\widetilde{w_i} \leftarrow \widetilde{w_i} +$ query result
17: {**Step 2: Monte Carlo estimation**}
18: $c \leftarrow 0$
19: $m \leftarrow Cn \log(1/\delta)/\varepsilon^2$
20: **for** $t \leftarrow 1, \ldots, m$ **do**
21:   Sample index $i$ with probability $\widetilde{w_i}/\sum_j \widetilde{w_j}$
22:   $z \leftarrow ()$
23:   **for** $u \leftarrow 1, \ldots, n$ **do**
24:     **for** $s^{(1)}, s^{(2)} \in [\ell]$ **do**
25:       $\widetilde{v_{s^{(1)},s^{(2)}}} \leftarrow$ the approximate probability that $(X_1, Y_1), \ldots, (X_n, Y_n)$ starts with $z, (s^{(1)}, s^{(2)})$, given that $(X_1, Y_1), \ldots, (X_n, Y_n)$ satisfies $E_i$     {*Using the approximate probabilistic inference oracle.*}
26:     Sample $s_1, s_2 \in [\ell]$ with probability $\widetilde{v_{s_1,s_2}}/\sum_{s^{(1)},s^{(2)} \in [\ell]} \widetilde{v_{s^{(1)},s^{(2)}}}$
27:     $z \leftarrow (z, (s_1, s_2))$
28:   **for** $j \leftarrow 1, \ldots, n$ **do**
29:     **if** $z$ satisfies event $E_j$ (i.e., $X_j \neq Y_j$ with prefix constraints) **then**
30:       **if** $i = j$ **then**
31:         $c \leftarrow c + 1$
32:       **break**
33: $\widetilde{Z_{b_1,\ldots,b_k}} \leftarrow (c/m) \sum_{i=1}^{n} \widetilde{w_i}$
34: **return** $\widetilde{Z_{b_1,\ldots,b_k}}$

---

Due to the fact that the inference oracle is only approximate, the above process introduces some errors that need to be handled.

## 3.2 APPROXIMATE SAMPLING

Algorithm 3 implements sequential sampling to generate samples from an approximate version $\widetilde{\pi}$ of the target distribution $\pi$. The algorithm builds the sample coordinate by coordinate, computing conditional probabilities at each step. The sampling procedure exemplifies the classical reduction from approximate sampling to approximate counting established by Jerrum et al. (1986). By estimating ratios of normalization constants, we convert the sampling problem into a sequence of approximate counting problems, each solvable using our normalization estimation subroutine.

The key insight is that the conditional probability $\pi(w_i = b | w_1, \ldots, w_{i-1})$ can be expressed as the ratio $Z_{w_1,\ldots,w_{i-1},b}/Z_{w_1,\ldots,w_{i-1}}$ (due to Claim 5). Note that both the numerator and denominator can be estimated using Algorithm 2.

The algorithm manages approximation errors by setting $\varepsilon_0 = \varepsilon/(2n)$ and $\delta_0 = \delta/(2n)$. This ensures that after $n$ sequential sampling steps, the total variation distance between the true distribution $\pi$ and the approximate distribution $\widetilde{\pi}$ is bounded by $\varepsilon$ with probability at least $1 - \delta$. The error analysis relies on the fact that approximation errors in conditional probabilities compound multiplicatively across coordinates.

---

**Algorithm 3** APPROXIMATESAMPLE: Sampling from $\widetilde{\pi}$ via approximate probabilistic inference.

---

**Require:** Bayes nets $P, Q$, parameters $\varepsilon, \delta$.
**Ensure:** Sample from distribution $\widetilde{\pi}$ that $(1 + \varepsilon)$-approximates $\pi$.
1: $\varepsilon_0 \leftarrow \varepsilon/(2n)$
2: $\delta_0 \leftarrow \delta/(2n)$
3: $\widetilde{Z} \leftarrow$ ESTIMATENORMALIZATION$(P, Q, (), \varepsilon_0, \delta_0)$
4: **for** $i \leftarrow 1, \ldots, n$ **do**
5:    {*Compute conditional distribution for position $i$.*}
6:    **for** $b \leftarrow 1, \ldots, \ell$ **do**
7:       $\widetilde{Z_{w_1,\ldots,w_{i-1},b}} \leftarrow$ ESTIMATENORMALIZATION$(P, Q, (w_1, \ldots, w_{i-1}, b), \varepsilon_0, \delta_0)$
8:       $\widetilde{\mu_b} \leftarrow \widetilde{Z_{w_1,\ldots,w_{i-1},b}}/\widetilde{Z}$
9:    Sample $w_i \leftarrow b$ with probability proportional to $\widetilde{\mu_b}$
10: **return** $w = (w_1, \ldots, w_n)$

---

### 3.3 CORRECTNESS ANALYSIS

**Theorem 6** (Main Result). *Algorithm 1 outputs $\widetilde{d_{\mathrm{TV}}}$, such that*

$$\mathbf{Pr}\left[ d_{\mathrm{TV}}(P, Q)/\left(1 + \varepsilon\right) \leq \widetilde{d_{\mathrm{TV}}} \leq (1 + \varepsilon) d_{\mathrm{TV}}(P, Q) \right] \geq 1 - \delta.$$

*Algorithm 1 makes $O(n^3 \ell^2 \varepsilon^{-2} \log \delta^{-1})$ calls to the approximate probabilistic inference oracle and runs in time $O(n^3 \ell \varepsilon^{-2} \log \delta^{-1})$.*

*Proof.* We prove the correctness of Algorithm 1. Note that Bhattacharyya et al. (2024) show that $Z \leq 2n \cdot d_{\mathrm{TV}}(P, Q)$. Therefore, $\widetilde{Z} \leq 2n \left(1 + \varepsilon\right) d_{\mathrm{TV}}(P, Q)$ with probability $1 - \delta/3$. Moreover, $Z/\left(1 + \varepsilon\right) \leq \widetilde{Z} \leq Z \left(1 + \varepsilon\right)$ with probability $1 - \delta/3$.

We have from the fact that $0 \leq f(w) \leq 1$ and Hoeffding's inequality (Lemma 3), that

$$\mathbf{Pr}\left[ \left| \frac{Z}{m\widetilde{Z}} \sum_{i=1}^{m} f(w^i) - \frac{Z}{\widetilde{Z}} \mathop{\mathbf{E}}_{\pi}[f(w)] \right| > \frac{\varepsilon}{\widetilde{Z}} d_{\mathrm{TV}}(P, Q) \right]$$

$$= \mathbf{Pr}\left[ \left| \frac{Z}{\widetilde{Z}} \sum_{i=1}^{m} f(w^i) - m \frac{Z}{\widetilde{Z}} \mathop{\mathbf{E}}_{\pi}[f(w)] \right| > \frac{m\varepsilon}{\widetilde{Z}} d_{\mathrm{TV}}(P, Q) \right]$$

$$= \mathbf{Pr}\left[ \left| \frac{Z}{\widetilde{Z}} \sum_{i=1}^{m} f(w^i) - m \mathop{\mathbf{E}}_{\pi}\left[ \frac{Z}{\widetilde{Z}} f(w) \right] \right| > \frac{m\varepsilon}{\widetilde{Z}} d_{\mathrm{TV}}(P, Q) \right]$$

$$\leq 2 \exp\left( -\frac{2m^2\varepsilon^2 d_{\mathrm{TV}}^2(P, Q)}{\widetilde{Z}^2 \sum_{i=1}^{m} \left( 0 - \frac{Z}{\widetilde{Z}} \right)^2} \right) = 2 \exp\left( -\frac{2m\varepsilon^2 d_{\mathrm{TV}}^2(P, Q)}{Z^2} \right) \leq 2 \exp\left( -\frac{m\varepsilon^2}{2n^2} \right),$$

which is at most $\delta/3$ whenever $m = \Omega\left( n^2 \varepsilon^{-2} \log \delta^{-1} \right)$. That is, $\left( Z/\widetilde{Z} \right) \left( \sum_{i=1}^{m} f(w^i) \right)/m$ is in the interval

$$\left( \mathop{\mathbf{E}}_{\pi}\left[ \frac{Z}{\widetilde{Z}} f(w) \right] - \frac{\varepsilon d_{\mathrm{TV}}(P, Q)}{\widetilde{Z}}, \mathop{\mathbf{E}}_{\pi}\left[ \frac{Z}{\widetilde{Z}} f(w) \right] + \frac{\varepsilon d_{\mathrm{TV}}(P, Q)}{\widetilde{Z}} \right)$$

with probability $1 - \delta/3$.

To finish the proof of correctness, we argue as follows. Note that $\widetilde{d_{\mathrm{TV}}} = \left( \widetilde{Z}/m \right) \sum_{i=1}^{m} f(w_i) = \left( \sum_{i=1}^{m} f(w_i)/m \right) \widetilde{Z}$. Therefore, with probability $1 - \delta$, we have that $\widetilde{d_{\mathrm{TV}}}$ is in the interval

$$\frac{\widetilde{Z}}{Z} \left( \mathop{\mathbf{E}}_{\pi} \left[ \frac{Z}{\widetilde{Z}} f(w) \right] - \frac{\varepsilon d_{\mathrm{TV}}(P, Q)}{\widetilde{Z}}, \mathop{\mathbf{E}}_{\pi} \left[ \frac{Z}{\widetilde{Z}} f(w) \right] + \frac{\varepsilon d_{\mathrm{TV}}(P, Q)}{\widetilde{Z}} \right) \widetilde{Z}.$$

By calculations, this interval becomes

$$\left( \mathop{\mathbf{E}}_{\pi}[f(w)] \, \widetilde{Z} - \varepsilon \frac{\widetilde{Z}}{Z} d_{\mathrm{TV}}(P, Q) \,, \mathop{\mathbf{E}}_{\pi}[f(w)] \, \widetilde{Z} + \varepsilon \frac{\widetilde{Z}}{Z} d_{\mathrm{TV}}(P, Q) \right),$$

or

$$\left( \frac{d_{\mathrm{TV}}(P, Q)}{Z} \widetilde{Z} - \varepsilon \frac{\widetilde{Z}}{Z} d_{\mathrm{TV}}(P, Q) \,, \frac{d_{\mathrm{TV}}(P, Q)}{Z} \widetilde{Z} + \varepsilon \frac{\widetilde{Z}}{Z} d_{\mathrm{TV}}(P, Q) \right),$$

or

$$((1 - \varepsilon) \, d_{\mathrm{TV}}(P, Q) - \varepsilon \, (1 + \varepsilon) \, d_{\mathrm{TV}}(P, Q) \,, (1 + \varepsilon) \, d_{\mathrm{TV}}(P, Q) + \varepsilon \, (1 + \varepsilon) \, d_{\mathrm{TV}}(P, Q)),$$

which is a subset of $((1 - 3\varepsilon) \, d_{\mathrm{TV}}(P, Q) \,, (1 + 3\varepsilon) \, d_{\mathrm{TV}}(P, Q))$. This concludes the discussion about the accuracy error. Note that the confidence error is $3\delta/3 = \delta$.

We now argue about the running time of Algorithm 1. The total running time of Algorithm 1 is the sum of the initial normalization estimate and the cost of the main sampling loop.

- The initial call to ESTIMATENORMALIZATION takes time polynomial in $n, \ell, 1/\varepsilon, \log(1/\delta)$.

- The main loop runs $m = O(n^2 \varepsilon^{-2} \log \delta^{-1})$ times.

- Inside the loop, each call to APPROXIMATESAMPLE involves an inner loop of $n$ steps. Each step makes $\ell + 1$ calls to ESTIMATENORMALIZATION. This dominates the cost of each iteration.

- The total number of calls to the inference oracle is therefore dictated by the $m$ samples, each requiring $O(n\ell)$ calls to ESTIMATENORMALIZATION, which itself makes queries. This leads to a total number of oracle calls of $O(n^3 \ell^2 \varepsilon^{-2} \log \delta^{-1})$. The overall running time is polynomial in all parameters.

This establishes that the algorithm is an FPRAS. $\qquad\square$

We may now argue about the correctness and the running time of Algorithm 2.

**Lemma 7.** *Algorithm 2 outputs* $\widetilde{Z_{b_1,\ldots,b_k}}$, *satisfying*

$$\mathbf{Pr}\left[ \left| \widetilde{Z_{b_1,\ldots,b_k}} - Z_{b_1,\ldots,b_k} \right| \leq \varepsilon Z_{b_1,\ldots,b_k} \right] \geq 1 - \delta.$$

*Moreover, the running time of Algorithm 2 is* $O\left(n^2 \ell^2 \log(1/\delta)/\varepsilon^2\right)$.

*Proof.* The algorithm estimates $Z_{b_1,\ldots,b_k} = \mathbf{Pr}[\bigcup_{i=1}^{n} E_i]$ where $E_i = (X_i \neq Y_i, X_1 = b_1, \ldots, X_k = b_k)$. First, we estimate each $w_i = \mathbf{Pr}[E_i]$ using approximate probabilistic inference queries. For $i \leq k$, we have that

$$\mathbf{Pr}[X_i \neq Y_i, X_1 = b_1, \ldots, X_k = b_k] = \mathbf{Pr}[(X_1, Y_1) \in S_1, \ldots, (X_n, Y_n) \in S_n]. \qquad (1)$$

In this case, for $j \leq k$ such that $j \neq i$, it is the case that $S_j = \{b_j\} \times [\ell]$, and $S_i = \{b_i\} \times ([\ell] \setminus \{b_i\})$. Moreover, $S_{k+1} = \cdots = S_n = [\ell]^2$.

For $i > k$, we have

$$\mathbf{Pr}[X_i \neq Y_i, X_1 = b_1, \ldots, X_k = b_k] = \sum_{b \in [\ell]} \mathbf{Pr}[b = X_i \neq Y_i, X_1 = b_1, \ldots, X_k = b_k]$$

$$= \sum_{b \in [\ell]} \mathbf{Pr}[(X_1, Y_1) \in S_1, \ldots, (X_n, Y_n) \in S_n]. \quad (2)$$

In this case, $S_1 = \cdots = S_k = \{b_j\} \times [\ell]$. Moreover, for $j > k$ such that $j \neq i$, it is the case that $S_j = [\ell]^2$, and $S_i = \{b\} \times ([\ell] \setminus \{b\})$. Each approximate query gives $\widetilde{w_i}$ such that $w_i/(1 + \varepsilon/n) \leq \widetilde{w_i} \leq w_i(1 + \varepsilon/n)$. This explains our reasoning behind Lines 1–16.

Line 21 samples an index $i$ proportional to its estimated weight $\widetilde{w_i}$. It follows that $i$ is sampled a probability that lies between $(1 - 2\varepsilon)w_i$ and $(1 + 2\varepsilon)w_i$.

The pseudocode in Lines 25–27 samples a $z$ conditioned on the event $E_i$ and has prefix $b_1, \ldots, b_k$. This procedure ensures that sample point $z \in E_i$, is sampled with probability proportional to $\mathbf{Pr}[z]/w_i$. Since this is done by making approximate inference calls, a $z \in E_i$ is sampled with probability that lies in the range $[(1 - 2\varepsilon)\mathbf{Pr}[z]/w_i, (1 + 2\varepsilon)\mathbf{Pr}[z]/w_i]$.

In Line 31 of Algorithm 2, counter $c$ is incremented only if $z$ does not belong to $E_1, \cdots E_{i-1}$. Thus, if $i$ is sampled in Line 21, then the probability that $c$ is increased in an iteration lies between $[(1 - 2\varepsilon)\mathbf{Pr}\left[E_i - \bigcup_{j=1}^{i-1} E_j\right]/w_i, (1 + 2\varepsilon)\mathbf{Pr}\left[E_i - \bigcup_{j=1}^{i-1} E_j\right]/w_i)$. Note that each $i$ is sampled with a probability between $(1 - 2\varepsilon)w_i$ and $(1 + 2\varepsilon)w_i$.

Putting it all together, we obtain that the probability that $c$ is incremented during an iteration is in the range $[(1 - 4\varepsilon)\mathbf{Pr}[E], (1 + 4\varepsilon)\mathbf{Pr}[E]]$, where $E = E_1 \cup \cdots \cup E_n$, which is the event $X \neq Y$. The estimator $\widetilde{Z_{b_1,\ldots,b_k}} = (c/m)\sum_{i=1}^{n} \widetilde{w_i}$ has expectation approximately $Z_{b_1,\ldots,b_k}$. By concentration bounds with $m = O(n\log(1/\delta)/\varepsilon^2)$ samples, the relative error is bounded by $\varepsilon$ with probability at least $1 - \delta$.

We shall now argue about the running time of Algorithm 2. The running time of Algorithm 2 is

$$O\left(mn\ell^2\right) = O\left(n^2\ell^2 \log(1/\delta_0)/\varepsilon_0^2\right) = O\left(n^2\ell^2 \log(1/\delta)/\varepsilon^2\right). \qquad \square$$

We now turn to Algorithm 3.

**Lemma 8.** *Algorithm 3 samples from distribution $\widetilde{\pi}$ where for all $w$*

$$\pi(w)/(1 + \varepsilon) \leq \widetilde{\pi}(w) \leq (1 + \varepsilon)\pi(w),$$

*with probability at least $1 - \delta$. The running time of Algorithm 3 is $O\left(n^3\ell^3 \log(1/\delta)/\varepsilon^2\right)$.*

*Proof.* Algorithm 3 samples from a distribution $\widetilde{\pi}$ iteratively, symbol-by-symbol. Assume that Algorithm 3 has sampled the first $k - 1$ symbols $w_1, \ldots, w_{k-1}$. To sample $w_k$, Algorithm 3 computes the approximate marginal

$$\widetilde{\mu_b} := \widetilde{\pi}(w_1, \ldots, w_{k-1}, b) = \frac{\sum_{w:(b_1,\ldots,b_k)=(w_1,\ldots,w_{k-1},b)} \widetilde{g}(w)}{\widetilde{Z}} = \frac{\widetilde{Z_{w_1,\ldots,w_{k-1},b}}}{\widetilde{Z}},$$

for every possible value $b \in [\ell]$, by using Algorithm 2. Then, Algorithm 3 samples $w_k$ proportional to the values $\{\widetilde{\mu_b}\}_{b=1}^{\ell}$. We will now bound the accuracy and confidence errors of Algorithm 3. Note that

$$\frac{Z_{w_1,\ldots,w_{k-1},b}}{Z(1 + \varepsilon_0)^2} = \frac{Z_{w_1,\ldots,w_{k-1},b}/(1 + \varepsilon_0)}{Z(1 + \varepsilon_0)}$$

$$\leq \widetilde{\mu_b} := \frac{\widetilde{Z_{w_1,\ldots,w_{k-1},b}}}{\widetilde{Z}} \leq \frac{Z_{w_1,\ldots,w_{k-1},b}(1 + \varepsilon_0)}{Z/(1 + \varepsilon_0)} = \frac{Z_{w_1,\ldots,w_{k-1},b}(1 + \varepsilon_0)^2}{Z},$$

with probability $(1 + \delta_0)^2$. Therefore, we may set $\varepsilon_0 := \varepsilon/(2n)$ so that $(1 + \varepsilon_0)^{2n} \leq 1 + \varepsilon$ and $\delta_0 := \delta/(2n)$ so that $(1 + \delta_0)^{2n} \leq 1 + \delta$, in order to satisfy the accuracy and confidence error requirements of Algorithm 3. The reason behind this choice, comes from the fact that (by calculations similar to the ones above) the actual marginal values $\mu_b$ satisfy $\mu_b = Z_{w_1,\ldots,w_{k-1},b}/Z$.

We shall now argue about the running time of Algorithm 3. Let $S(n)$ be the number of steps to sample $n$ symbols from $\widetilde{\pi}$. Algorithm 3 gives the recurrence relation $S(n) = O(n^2\ell^3 \log(1/\delta)/\varepsilon^2) + S(n - 1)$, since Algorithm 3 invokes Algorithm 2 exactly $\ell$ times for every symbol it samples. The latter yields $S(n) = O\left(n^3\ell^3 \log(1/\delta)/\varepsilon^2\right)$. $\qquad \square$

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
