# OpenReview forum: "Approximate Inference Suffices for Statistical Distance Estimation"
_ICLR.cc/2026/Conference — Submitted to ICLR 2026_

### Official Review · Reviewer_DM9u · 2025-10-24

**Soundness:** 3
**Presentation:** 1
**Contribution:** 3
**Rating:** 8
**Confidence:** 2

**Summary:**

The authors consider the problem of estimating statistical distance between two Bayes nets using approximate probabilistic inference queries. Roughly, the problem is defined as follows.

First, the probability distribution of a Bayes net is as follows: Given a directed graph and a collection of conditional probability tables, the probability of a labeling of the vertices is given by the product of the conditional probabilities of all vertex labels.
The statistical distance between two distributions is the $\ell_1$ distance between two distributions viewed as vectors.
The goal is to compute statistical distance using approximate probabilistic inference queries: for any arbitrary sets S_1, …, S_n, compute (up to approximate multiplicative error), the probability $X_i \in S_i$ for all $I$.

The general approach of previous works was to define an estimator $f$ and distribution $\pi$ such that the expected value of $f$ over $\pi$ is the statistical distance (possibly normalized by some easy to compute constant $Z$). The authors compute an auxiliary Bayes net where the labels are distributed over a joint distribution $(X, Y)$ with $X \sim P, Y \sim Q$. The two key steps are then to construct the appropriate $f, \pi$ and estimate $Z$. In this work, the authors show how to sample from and estimate parameters of the relevant distributions using approximate probabilistic inference oracles, rather than exact oracles.

Conclusion

Overall, the paper studies an interesting question and finds a nice reduction between two well studied problems. For this reason, I tend towards accept. However, as I am not a domain expert, I found the paper a bit hard to follow (see below comment on usage of prior work) and thus assign a low confidence score. While I did not check the details very closely, the claims seem reasonable and the proofs I did check seem correct.

**Strengths:**

The paper considers the natural problem of estimating statistical distance in Bayes nets. They establish an interesting result, that statistical distance can be estimated with approximate probabilistic inference queries. This can be viewed both positively and negatively: 1) hardness results in statistical distance estimation can imply that probabilistic inference is hard, or 2) algorithms for probabilistic inference can directly by applied to statistical distance estimation.

**Weaknesses:**

Perhaps due to the fact that I am not very familiar with the techniques in the paper, the algorithm are a bit hard to follow. Many details seem to be elided over: 1) what is importance sampling? (Yes a standard technique, but good to define and motivate). 2) How do you “carefully” define the CPT to ensure the condition required in 152? 3) Why is $Z = \Pr(X \neq Y)$? And this seems crucial so it is strange there is no justification for this. More generally, it feels that this paper builds off Bhattacharya et al 2024 in a significant way, but does not do so in a self-contained way, assuming the reader has read the prior work.

**Questions:**

It seems to me that the joint distribution $(X, Y)$ is supposed mimic coupling such that $\Pr(X \neq Y) \sim TVD(P, Q)$ i.e. the optimal coupling. If this is the case, why not just estimate $Z$?

Minor Comments

110 - should running time of FPRAS by polynomial in $\log(1/\delta)$?

---

> ### Author Response · Authors · 2025-11-21
>
> Dear Reviewer,
>
> Thank you for your feedback.
>
> Let us address your concerns and questions below.
>
> (1) We have added an explanation of importance sampling in our rebuttal revision.
>
> (2) Please note that we are enforcing this condition (now in Line 169).
> We give additional details of how to construct the CPTs in Section 2.2.
>
> (3) We have added in our rebuttal revision a derivation of the fact that $Z = \Pr[X \neq Y]$.
>
> (4) The joint distribution does not mimic optimal coupling but mimics a local coupling.
> In case of optimal coupling, $\Pr[X \neq Y]$ is indeed equal to $d_{TV}(P,Q)$.
> However, by using a local "partial coupling," we are defining $Z = \Pr[X \neq Y]$ as stated in Lines 163--164, so that $d_{\rm TV}(P, Q) = Z \cdot \mathbb{E}_{w \sim \pi} [f (w)]$, for an appropriately defined estimator function $f$ and distribution $\pi$.
> Thus, simply estimating $Z$ does not suffice; we need to be able to sample efficiently from $\pi$.
>
> (5) This is correct about $\log(1 / \delta)$, thank you for pointing it out.
> Our FPRAS indeed runs in time polynomial in $\log(1 / \delta)$.

---

> ### Comment · Reviewer_DM9u · 2025-11-25
>
> Thank you for the clarification and the rebuttal. I remain positive about the paper.

---

### Official Review · Reviewer_ZwwX · 2025-10-27

**Soundness:** 2
**Presentation:** 2
**Contribution:** 2
**Rating:** 4
**Confidence:** 2

**Summary:**

The authors prove that estimating total variation distance between probability distributions over the same underlying directed acyclic graph (DAG) can be reduced to approximate probabilistic inference while previous work by Bhattacharyya et al. (2024) showed this reduction was possible with exact probabilistic inference.
The main idea is to perform normalization estimation, approximate sampling and Monte Carlo estimation and hence construct the reduction by designing an FPRAS (Fully Polynomial-time Randomized Approximation Scheme).
The authors show that one only needs $O(n^3\ell^2\epsilon^{-2} \log \delta^{-1})$ approximate inference queries where $n$ is the number of nodes in the DAG and $\ell$ is the range of the random variables to approximate the total variation distance up to an $(1\pm \epsilon)$ factor with probability $1-\delta$.

**Strengths:**

- The result extends the previous limitation that such a reduction can be done with exact probabilistic inference.

**Weaknesses:**

- The result may be incremental that the main idea is from the previous work (Bhattacharyya et al.)

**Questions:**

- Line 137: I am not sure what we are normalizing with $Z$.

- Line 144: It may be helpful to mention that $g$ is the surrogate function for $g^\ast$

---

> ### Author Response · Authors · 2025-11-21
>
> Dear Reviewer,
>
> Thank you for your feedback.
>
> Let us address your questions below.
>
> (1) At this point in our paper it is not clear why $Z$ is called a normalization constant.
> However, when we define $\pi$, it is apparent that $Z$ can be viewed as a normalization constant.
>
> (2) Thank you for pointing this out.
> We have mentioned in our rebuttal revision that $g$ is a surrogate function for $g^*$.

---

### Official Review · Reviewer_MD4a · 2025-10-28

**Soundness:** 2
**Presentation:** 1
**Contribution:** 2
**Rating:** 4
**Confidence:** 2

**Summary:**

The paper studies the computational relationship between two fundamental problems—statistical distance estimation (specifically total variation distance) and probabilistic inference in Bayesian networks. Building on recent work (Bhattacharyya et al., 2024), which showed a reduction from statistical distance estimation to exact probabilistic inference, this paper proposes a reduction to approximate probabilistic inference instead. The main claim is that there exists an FPRAS for estimating total variation distance using approximate inference queries, potentially broadening the applicability of such methods to practical settings where exact inference is infeasible.

**Strengths:**

- The paper targets an important and fundamental question in computational learning theory, i.e., linking statistical distance estimation and probabilistic inference

**Weaknesses:**

- The paper is entirely theoretical with no numerical experiments or case studies to demonstrate how the proposed algorithms might perform in realistic scenarios.

- While technically detailed, the paper is difficult to follow. Algorithmic steps (especially in Algorithms 2 and 3) are densely described without clear intuition or illustrative examples. The exposition would benefit from diagrams or toy examples linking the theory to intuitive settings.

**Questions:**

N/A

---

> ### Author Response · Authors · 2025-11-21
>
> Dear Reviewer,
>
> Thank you for your feedback.
>
> Thank you for highlighting the need for expanding on intuition.
> We have expanded the description of Algorithms 2 and 3 in our rebuttal revision to provide more intuition. Please let us know what more can be done on our end for you to increase the score.

---

### Official Review · Reviewer_pxw5 · 2025-10-30

**Soundness:** 3
**Presentation:** 3
**Contribution:** 3
**Rating:** 6
**Confidence:** 4

**Summary:**

In this paper, the authors study the problem of computing the total variation distance between two probability distributions $P$ and $Q$, each represented as Bayesian networks defined on the same directed acyclic graph (DAG) $G$. The objective is to estimate the total variation distance $d_{TV} (P,Q)$ between these two distributions.

In their ICML 2024 paper, Bhattacharyya et al. established a key connection between total variation distance estimation and probabilistic inference. They showed that if one has access to an exact probabilistic inference oracle, the total variation distance between two Bayes nets can be computed in polynomial time via an exact reduction. However, this reduction critically depends on exact computation of certain normalization constants such as $Z=Pr⁡[X\neq Y]=1-Pr⁡[X=Y]$. When only approximate inference is available, multiplicative errors in estimating $Pr⁡[X=Y]$ can amplify dramatically in its complement $1-Pr⁡[X=Y]$, especially when $Pr⁡[X=Y]$ is close to $1$. As a result, even small relative errors in inference can lead to unbounded relative errors in the estimated total variation distance.

The present paper overcomes this instability by introducing a new, structure-preserving randomized reduction that avoids direct dependence on such complements. It demonstrates that access to a $(1+\varepsilon)$–relative approximate inference oracle suffices to obtain a $(1+\varepsilon)$–approximation of $d_{TV}(P,Q). This establishes the first reduction from total variation distance estimation to approximate probabilistic inference, thereby strengthening the algorithmic connection between the two problems.

To obtain this result, the authors adapt the classical Jerrum–Valiant–Vazirani paradigm that connects approximate counting and sampling, extending it to the setting of Bayesian networks. They develop an importance-sampling framework that estimates normalization constants and draws approximate samples using only approximate inference queries, while carefully controlling the propagation of multiplicative errors.

**Strengths:**

The paper makes an interesting contribution by strengthening the connection between two classical and well-studied problems in machine learning, namely, statistical distance estimation and probabilistic inference. The technical approach appears non-trivial.

**Weaknesses:**

My main concern is that the result feels somewhat incremental relative to prior work. Since both exact and relative approximate inference, as well as total variation distance estimation, are known to be #P-hard, the strengthened reduction does not yield new tractable cases. Consequently, the practical algorithmic implications of this improved connection appear limited.

**Questions:**

Are there specific classes of models or structural assumptions under which approximate inference is known to be tractable?

---

> ### Author Response · Authors · 2025-11-21
>
> Dear Reviewer,
>
> Thank you for your feedback.
>
> Regarding your main question and concern, please take note that approximate inference is a well-studied topic in Bayesian networks with a long series of results from theory as well as practice. Practical methods such as variational inference and belief propagation have been shown to converge even when the exact inference is intractable in practice.
> Theoretical results have often shown tractability under conditions on the CPT parameters [1, 2, 3, 4] --- these methods have also been shown to converge in practice.
>
> The practical implication of our result is that for the classes of networks for which approximation techniques such as variational inference and belief propagation converge, it is also possible to approximate the total variation distance.
>
> [1] https://dl.acm.org/doi/10.5555/3009657.3009733
>
> [2] https://arxiv.org/pdf/1301.7392
>
> [3] https://arxiv.org/pdf/1105.5462
>
> [4] https://people.eecs.berkeley.edu/~jordan/papers/variational-intro.pdf
>
> We have added a discussion about instances where approximate inference is possible (see Section 1).

---

> > ### Comment · Reviewer_pxw5 · 2025-11-25
> > **clarification**
> >
> > As far as I can tell from reading the references you cite, they provide asymptotic convergence guarantees under specific parameter assumptions. However, they do not appear to establish the existence of a polynomial-time algorithm that, for any $\varepsilon > 0$, returns a $(1+\varepsilon)$-multiplicative approximation in time polynomial in the network size and $1/\varepsilon$. Because your FPRAS for TV distance explicitly requires access to such an oracle, it is not clear to me how the reduction can yield a polynomial-time algorithm for total variation distance computation in any concrete class of Bayesian networks. I might be missing something and would appreciate a clarification.

---

> ### Author Response · Authors · 2025-11-26
>
> Thank you for your response. We think there might be a slight misunderstanding as to what our result implies. Our result shows that statistical distance (TV distance) can be approximated in polynomial time given access to an approximate inference oracle. Therefore, if the underlying inference oracle is polynomial time for a class of networks, then we can derive that statistical distance is polynomial time. It is important to note that the algorithm for estimation of statistical distance doesn't require the inference oracle to be in polynomial time, neither does it require the underlying algorithm to be randomized—it admits methods that will return an answer with (1+ε) error after converging.
>
> What is more significant is that our result allows one to use an approximate inference oracle: for example, consider a class of Bayesian networks for which there exists an approximate inference technique that always converges; then we can run that inference technique until the lower and upper bounds are within a factor of (1+ε) and return those as answers to inference queries.
>
> Finding a clean structural characterization of Bayesian networks that admit polynomial-time inference is, of course, notoriously hard, but that is a reflection of the limitation of theoretical tools, not the limitation of algorithms which are shown to converge in practice, and for many of these algorithms, we have asymptotic convergence guarantees. Therefore, what our results imply is that for all such classes, it is now practically possible to estimate statistical distance.
>
> Note that the prior work of Bhattacharyya et al. (2024) didn't allow approximate inference oracles, so we couldn't return such  approximate methods.
>
> Please let us know if any further clarification is needed and in case there is no need for clarification, we hope you would consider revising your score.

---

### Official Review · Reviewer_qjsT · 2025-11-09

**Soundness:** 3
**Presentation:** 2
**Contribution:** 3
**Rating:** 6
**Confidence:** 3

**Summary:**

This paper studies the relation between learning total variation distance and probabilistic inference in Bayesian nets. The authors show that total variation distance can be approximated in polynomial time using an approximate probabilistic oracle for Bayesian nets.

**Strengths:**

The authors establish a strong theoretical connection between probabilistic inference and estimating total variation distance in Bayesian nets. The authors prove that it is possible approximate the total variation distance between Bayes nets in fully polynomial time using approximate oracles for probabilistic inference, which estimates the probability of any given event. This improves upon the work of Bhattacharya et.al. 2024 which required exact probabilistic inference oracles.

**Weaknesses:**

The writing and presentation needs improvement. While many results are developed from the previous work, Bhattacharya et.al. 2024, it is important to make the proof and statements self-contained.

**Questions:**

1. Are there time complexity lower bounds for the reduction? I guess a $\Omega(n^2)$ lower bound is trivial which is the time to write down the nodes and edges. Could one argue something better than this?

---

> ### Author Response · Authors · 2025-11-21
>
> Dear Reviewer,
>
> Thank you for your feedback.
>
> We have striven to improve some key parts of the presentation.
> Please have a look at our rebuttal revision (especially at the edited parts highlighted in blue color) and let us know if something needs to be improved further.
>
> Regarding your question, we interpret it in two parts, as follows:
>
> (a) Regarding the size of the reduced instance.
> Regarding this, the reduction is optimal in a certain sense: the reduction is structure preserving in the sense that the reduced Bayes net graph is the same as the original input Bayes net graph.
> However, the size of the alphabet is doubled, and hence the size of the CPT table will only be increased by a constant factor.
>
> (b) Regarding the number of approximate inference queries to the reduced instance.
> Our algorithm makes $\widetilde{O}(n^3 \ell^2 \log(1 / \delta) / \varepsilon^2)$ queries, in the worst case (see Theorem 5 in Section 3.3).
> It is a very interesting question whether we can reduce the number of queries.
> However, establishing a lower bound on the number of queries will be challenging, as time-complexity lower bounds are difficult to prove.
> Even establishing that more than one query is needed is a challenging task.

---

> > ### Comment · Reviewer_qjsT · 2025-11-25
> >
> > Thanks for your response. I see that the presentation has indeed improved. My evaluation remains unchanged.

---

### Meta-Review · Area_Chair_iE3C · 2026-01-05

**Summary:**

This paper proves that estimating total variation distance between probability distributions over Bayesian networks can be reduced to approximate probabilistic inference, building on recent work by Bhattacharyya et al. (2024) that showed a reduction to exact probabilistic inference. While the paper contains valuable contributions, the reviewers also raised concern that the result is incremental relative to prior work. Given the highly competitive nature of this conference, I thus consider this a borderline paper, and finally decided for rejection.

**Reviewer Concerns:**

Concerns addressed during rebuttal:
- Reviewers qjsT, MD4a, and DM9u pointed out that writing and presentation needs improvement. Specifically, many results are developed from Bhattacharya et.al. (2024) which should be made self-contained, and the algorithms are difficult to follow without examples or clear intuition. The authors have updated the manuscript to address these concerns.

Concerns that are not fully addressed:
- Reviewers pxw5 and ZwwX raised concern that the result is incremental relative to prior work.
- Reviewer MD4a pointed out that the paper has no numerical experiments or case studies to demonstrate how the proposed algorithms might perform in realistic scenarios.

**Reviewer Scores:**

Reviewers may have maintained their scores.

---

### Decision · Program_Chairs · 2026-01-26

Reject